# The Usefulness of Lymphadenectomy in Bladder Cancer—Current Status

**DOI:** 10.3390/medicina57050415

**Published:** 2021-04-25

**Authors:** Bartosz Małkiewicz, Paweł Kiełb, Adam Gurwin, Klaudia Knecht, Karol Wilk, Jakub Dobruch, Romuald Zdrojowy

**Affiliations:** 1Department of Urology and Oncologic Urology, Wroclaw Medical University, 50-556 Wroclaw, Poland; pk.kielb@gmail.com (P.K.); gurwin.adam@gmail.com (A.G.); klaudia.knecht@gmail.com (K.K.); karolwilk@me.com (K.W.); romuald.zdrojowy@umed.wroc.pl (R.Z.); 2First Department of Urology, Centre of Postgraduate Medical Education, 01-813 Warsaw, Poland; kubadobr@gmail.com

**Keywords:** bladder cancer, surgery, lymph node dissection, lymphadenectomy, staging

## Abstract

The purpose of this review is to present the current status of lymph node dissection (LND) during radical cystectomy in patients with bladder cancer (BCa). Despite the growing body of evidence of LND utility at the time of radical cystectomy (RC) in high-risk nonmuscle-invasive and muscle-invasive BCa (MIBC), therapeutic and prognostic value and optimal extent of LND remain unsolved issues. Recently published results of the first prospective, a randomized trial assessing the therapeutic benefit of extended versus limited LND during RC, failed to demonstrate survival improvement with the extended template. Although LND is the most accurate staging procedure, the direct therapeutic effect is still not evident from the current literature, limiting the possibility of establishing clear recommendations. This indicates the need for robust and adequately powered clinical trials.

## 1. Introduction

Treatment of advanced bladder cancer is complex. It mainly consists of neoadjuvant chemotherapy and radical cystectomy combined with lymphadenectomy. The management of the lymph nodes (LN) deserves special attention, as nodal metastases, apart from the local advancement, are the most important prognostic factor in patients with bladder cancer. The LN invasion has negative impact on the cancer-specific survival and overall survival, even without other unfavorable prognostic factors. In a study evaluating 1100 patients undergoing radical cystectomy, the five-year relapse-free survival, regardless of the degree of advancement of pT, for pNo, pN1, pN2, and pN3, was 81.4%, 29.3%, 18.2%, and 0%, respectively [1].

The current guidelines for LND in bladder cancer are imprecise and differ in recommendations. In this review, we summarize the current medical knowledge and analyze several retrospective and nonrandomized, prospective studies presenting individual results of lymphadenectomy. In addition, we cite results of recently prospective randomized trials that compare the oncological effect of limited vs. superextended lymphadenectomy, and entirely new, promising treatments for bladder cancer.

## 2. Imaging and Molecular Diagnosis of Nodal Metastases

The most commonly used imaging method assessing regional staging in BCa is abdomino-pelvic CT scan [2,3,4]. Although in previous versions of the TNM classification, the size of the lymph node was considered to be a marker of nodal invasion and was the basis for the classification of N feature, it is now considered that the size criterion as the single parameter is imprecise, due to the wide range of sizes of normal LNs and possible deposits of cancer cells in unenlarged nodes [5,6]. Currently, short node axis limits of at least 8 mm are recommended for the pelvic area and 10 mm for the abdomen [3,7]. A circular or irregular node contour suggests metastatic changes [6,8]. An example of an internal nodal architecture disorder is the low central attenuation caused by necrosis [9]. Studies published over the last 10 years have shown that even with improved technology, CT sensitivity in detecting LN metastases varies within the wide range from 30% to 75%. This is probably mainly due to micrometastases in normal-sized lymph nodes [10,11,12]. In the same literature, the CT specificity values ranged from 56% to 100%, while accuracy was assessed between 61% and 97%. In general, CT in detecting involved LN can be seen as a more specific method than sensitive [13].

An increasingly common alternative to CT scans for the diagnosis of nodal metastases is MRI [3]. The criteria and limitations for lymph node size are similar to those for CT [8,13,14]. Compared to tomography, the use of multiple pulse sequences in MRI enables evaluating many additional parameters. Metastatic lymph nodes are more likely to exhibit heterogeneous and/or increased peripheral amplification and centrally located hypointense areas in phase T1 and hyperintense in phase T2, which correspond to foci of necrosis [9,15]. In a meta-analysis conducted by Woo et al., the MRI sensitivity in the tested aspect was set at 56%, even though the specificity was 94% [16]. In another study, MRI sensitivity and specificity ranged from 40.7% to 86% and 31% to 92%, respectively [15]. One form of improvement in MRI diagnostic parameters is imaging after intravenous administration of ultrasmall iron oxide molecules with paramagnetic characteristics—USPIO (ultrasmall superparamagnetic particles of iron oxide) with lymphotropic features [17,18]. In normal nodes, USPIO is captured, which in MRI is characterized by spilled hypointensity in T2 and gradient-echo sequence in this imaging [6,19]. In metastatic nodes, the tumor areas in the same sequence remain hypertensive. The use of USPIO increased sensitivity from 56% to 86% with unchanged specificity, as demonstrated in several analyses [16,17]. It should be emphasized that these parameters apply to unenlarged nodes but already with deposits of cancer cells. Subsequent studies have sought to remove the limitations of this method. The modification of the USPIO–DW–MRI reduced the number of sequences needed for analysis and eliminated the need to compare parameters for individual nodes [20].

Positron emission tomography (PET/CT) is another imaging technique which, according to some radiological societies, may be suitable for estimating the pathological stage according to the TNM classification with limited utility for the assessment of T-characteristics due to physiological excretion of the radiomarker fluorodeoxyglucose (FDG) through the urinary tract [13,21]. Metastatic LN is characterized by increased metabolic activity, thereby exhibiting more intense capture of FDG in PET/CT. Unfortunately, PET/CT has a lower accuracy in detecting metastases in unenlarged (<5 mm) LN [6]. In addition, increased marker capture can be observed in inflammatory or reactive lymph nodes. As in the case of CT and MRI, the PET/CT examination has a limited sensitivity and a much higher level of specificity, which in the meta-analysis ranged between 33% and 78% for sensitivity and 86.7% and 100% for specificity [22], respectively. Compared to the conventional CT scan, the PET/CT method using FDG showed a sensitivity of 69% for the PET/CT study and 45% for CT scans; the specificity values were 95% and 98%, respectively [23]. The use of radioactive carbon-labeled choline did not show better diagnostic parameters in the LN assessment (sensitivity 66%, specificity 89%) [24].

Limitations of imaging diagnostics lead to the search for other possibilities for preoperative determination of metastases in regional lymph nodes. Mitra et al. used the real-time polymerase chain reaction (RT-PCR) technique to determine a panel of 70 genes involved in critical oncogenesis pathways. The authors concluded that the calculation considering the combination of determinations for the ICAM, MAP2K6, and KDR genes allows for 90% accuracy in predicting patients with positive lymph nodes [25]. Other techniques using RT-PCR focusing on different cancer-specific genes, such as CK20, UPII, and EGFR, have also been described [26]. Flow cytometry and immunocytochemical analysis were evaluated to diagnose cancer cells presence in serum based on the identification of epithelial cell adhesion molecule (EpCAM), cytokeratin, and nonexpression of CD45 and DAPI [27]. The authors identified cancer cells in 57.1% of patients with metastatic bladder cancer. A comprehensive meta-analysis by Msaouel and Koutsilieris showed correlations between cancer cell presence in the bloodstream and the more advanced cancer stage [28]. Concerning urothelial cancer, the relationship between stage and cancer markers such as CEA and CA19-9 is described. However, these results are controversial and still far from clinical use [29]. The serum matrix metalloproteinase-7 level (MMP7) assessment showed a significant correlation with nodal metastases in patients with bladder cancer with sensitivity and specificity of 82% and 71%, respectively, at serum concentrations 7.15 ng/mL [30,31].

Diagnostic limitations of imaging methods combined with the lack of widespread use of molecular techniques make lymphadenectomy as the most valuable method in detecting regional lymphatic spread in bladder cancer patients.

## 3. Lymphadenectomy in Bladder Cancer

In the 1830s, Colston and Leadbetter demonstrated that pelvic metastases in bladder cancer patients were surgically resectable [32]. In 1946, Jewett and Strong described the pelvic area as a “cardinal metastasis site,” later establishing the first commonly accepted bladder cancer assessment system: Jewett–Marshall [33,34]. It was only when, in 1982, Skinner described long-term survival in patients with metastases to lymph nodes undergoing pelvic lymphadenectomy that this procedure began to be considered standard [35]. Moreover, radical cystectomy alone has been shown not to provide optimal therapeutic results [36].

### 3.1. Lymphadectomy Template—Anatomical Considerations

Leadbetter and Cooper in 1950 described the anatomy of the lymphatic drainage of the bladder, which can be divided into six different areas: (1) visceral lymphatic plexus in the bladder wall; (2) intermediate lymph nodes within the perivesical fat arranged in the anterior, lateral and posterior groups; (3) pelvic lymphatic trunks, located medially from the groups of external and internal iliac nodes; (4) regional pelvic lymph nodes, comprising external and internal and presacral iliac lymph nodes; (5) lymphatic trunks carrying lymph from regional pelvic lymph nodes; and (6) common hip nodes [37,38]. Anatomical studies have also helped to define the different levels of drainage [39]. The original drainage begins with the areas of the external and internal iliac vessels and the obturator fossa. Secondary drainage comes from the area of the common iliac vessels, while the tertiary flow concerns the trig one area and the posterior bladder and is located in the presacral nodes [40]. Despite numerous attempts to define, the range of lymphadenectomy is constantly discussed [41]. In 2004, Leissner et al. proposed a three-level LND scheme. Level I extends proximally to the level of division of the common iliac artery, the nodes included in the level II template are located laterally from the common iliac vessels and proximal to the aortic bifurcation. The nodes in the Level III template are located between the ureter and the aorta up to the lower mesenteric artery (Figure 1) [42].

Many studies use a system consisting of templates called limited (Figure 2a), standard (Figure 2b), extended (Figure 2c), and superextended (Figure 2d). The limited LND usually covers both sides of obturator fossa area (Figure 2a) [43,44]. The standard range of LND includes the removal of nodes in the area determined by the following boundaries: proximal—division of the common iliac artery; distal—inguinal ligament; lateral—femoral; and medial nerve—bladder wall (Figure 2b) [43,45]. The extended template refers to the resection of nodes located between the aortic bifurcation and the iliac vessels (proximal), the genital–femoral nerve laterally, the iliac ocular vein (distal), and the internal iliac vessels (rear) (Figure 2c) [41]. The superextended area covers these areas, but the proximal limit is located at the level of the inferior mesenteric artery (Figure 2d) [42,44].

Many authors stressed the need for a bilateral lymphadenectomy. Abol-Enein et al. in the paper defining the metastatic topography, demonstrated their bilaterality in 39% of cases [46]. In a study mapping metastasis using SPECT–CT and intraoperative γ-probe, it was found that the involved LN were in 15% of cases contralateral to the primary tumor location [47]. According to the authors’ classification, when metastases were found at the level I, positive nodes were also identified in 57% and 31% of cases at levels II and III, respectively. If level II metastases were included, in 35% of cases, they were also at Level III. Positive nodes at level III were found only when metastases were present at the lower ones, confirming that the pelvic region is the primary anatomical spread site. If a LND were performed in the standard range, metastases would be overlooked in 6.8% of patients [42]. These findings were confirmed in subsequent reports, which concluded that up to 41% of metastatic lymph nodes are outside the standard LND template [40,45,48]. It was essential to include the presacral area in a routine lymphadenectomy. It showed metastasis in 30% of patients if common iliac nodes were occupied. The same analysis found that patients with locally advanced disease (>pT2) in 16% of cases had nodal metastases outside the standard lymphadenectomy area [49].

These data confirm some crucial facts. First, the bladder’s lymphatic drainage seems bilateral, which contradicts unilateral LN removal, even if the primary tumor is one-sided. Second, most of the primary lymphatic drainage appears to be within the pelvis. It should be borne in mind, however, that the potential drainage is more extended and a small but attention-requiring part of the nodal metastases is outside the pelvic boundaries, which is an argument for a broader range of lymphadenectomy [50].

### 3.2. Anatomical Boundaries and Oncological Results

Data on the fundamental therapeutic role of lymphadenectomy in cystectomy patients remain controversial. In an analysis of 1091 cystectomies over three years, limited, standard, and extended lymphadenectomies were performed in 144 (13%), 729 (67%), and 101 (9%) of cases. In 117 (11%) patients, they were not performed at all [51]. The available data highlight the oncological benefits of nodal resection during cystectomy compared to its absence [52,53,54,55,56,57,58]. Statistically significant reductions in overall (36% vs. 45%) and cancer-specific (54% vs. 65%) mortality were observed during five-year observation [59]. Bruins et al., in a systematic review, identified seven independent cohorts in which patients who had a LND had better oncological results than patients with similar disease characteristics who had only cystectomy [60]. Despite this improvement in results, there are controversies regarding application of LND in patients with cN0 disease, resulting from prolonged surgery time. However, it should be remembered that the cystectomy itself is the greatest risk factor for complications. Node-free patients have also been shown to benefit oncologically after LND [61]. May et al. assessed 1291 patients after radical cystectomy and PLND without nodal metastases on a histopathological examination. Removal of >16 LNs reduced the likelihood of dying from cancer [62]. Undoubtedly, the removal of possible micrometastases through the use of extended lymphadenectomy template has oncological benefits. However, this association may also be partly the result of the Will Rogers phenomenon, where better disease identification leads to the transfer of patients from a healthy human population to a diseased group. Thus, patients who would be classified as pN0 with a limited LND procedure were reassigned to pN+ after testing a large number of nodes resulting from the extended boundaries [63,64].

Limited lymphadenectomy is associated with a small number of pathologically assessed lymph nodes, so there is inadequate information about the advancement of the disease [65,66]. In addition, it was shown that progression-free survival, disease-specific survival, and overall survival in the LND-limited group were statistically lower compared to the second studied population [44]. The most common standard lymphadenectomy is the area in which the greatest number of metastases is located, and the average number of removed nodes in this area is 13 (range 9–18) [34,42,51,67]. In recent decades, the results of standard vs. extended LND have been compared many times [48,52,68,69,70]. The largest retrospective analysis, involving 658 patients in the extended group, reported a higher proportion of pN+ patients (26% vs. 13%) and an improvement in five-year progression-free survival (35% vs. 7%) [69]. A meta-analysis of six studies comparing both LND templates showed a positive effect of the extended procedure on progression-free survival in the pN0 group (HR 0.68, 95% CI 0.51–0.90), pN+ (HR 0.58, 95% CI 0.47–0.72), and in pT3-4 patients (HR 0.61, 95% CI 0.52–0.73) [41]. In theory, a willingness to accurately assess node involvement and control cancer locally would support an even wider range of LND. Several studies have been published evaluating the efficacy of superexpanded lymphadenectomy [71,72,73]. Neither of these had benefits in relapse-free, complete, and cancer-specific survival. The results concerned both pN0 and pN+ patients. In addition, the long-awaited results of the first prospective randomized trial showed no benefit from such an extensive lymphadenectomy. The five-year relapse-free survival was higher in the extended group (64.6%) compared to the limited LND group (59.2%), but the difference (5.45% (95% CI–6.43% to 17.33%)) did not reach statistical significance (hazard ratio 0.84 (95% CI 0.58–1.22); *p* = 0.36). Similar results were obtained in assessing overall and disease-specific survival [74,75]. The lack of oncological benefit may be due to the fact that the presence of metastases outside the pelvic region is associated with a higher risk of visceral metastases.

### 3.3. Prognostic Interpretation of N+

The latest TNM classification on which basis the stage of bladder cancer is assessed was developed in 2017 and is widely used by both oncologists and urologists [76]. However, this system has many limitations. It does not consider essential parameters such as the quality and extent of LND, the total number of removed lymph nodes, the number of positive lymph nodes, the size of metastases, or the ratio of positive to total lymph nodes.

It is well known that removing more nodes can improve survival [77,78,79,80,81]. In all studies, survival improved in relation to the number of removed LNs, and this trend was found to be independent of nodal status. Lower death risk was observed in patients who had at least 10–14 lymph nodes removed [82]. In two independent multicenter studies, the authors proposed a minimum number of nodes of 25, which assures the absence of lymphatic metastases [83,84].

The increasing number of positive nodes is reflected in lower overall survival. Abdel-Latif et al. demonstrated that the median three-year survival in patients with one, two, to five and more positive nodes was 58.6%, 31.8%, and 6.8%, respectively [85]. Similar results were observed using the increasing cutoff values for the four, five, and six positive LNs [81,86,87]. Bruins et al. published an analysis of 369 pN+ patients and showed better results in patients with two or fewer metastatic lymph nodes, achieving a five-year relapse-free survival of 44% compared to 24% in the group with LN ≥ 2 [74]. The studies suggest that if the number of positive LNs is in the range of 1 to 4, each additional positive lymph node worsens survival. With five or more LN+, the mass of metastases is so significant that an additional positive node does not alter the clinical outcome.

The fact that the chances of finding one positive LN in a more extensive pelvic lymphadenectomy were greater than in a restricted procedure prompted researchers to postulate that the significance of LN+ disease would be different in patients with different numbers of removed nodes [69]. To solve this problem, the lymph node density (LD) concept was proposed as a prognostic tool to evaluate the ratio between LN positive and total lymph nodes removed to stratify patient prognosis [57]. Herr et al. found that the five-year overall survival was reduced from 64% to 8% when LD was >20%; the threshold was established by multivariate analysis. Other publications subsequently corroborated these findings, and although some authors used different cutoff values, the 20% limit remains the most widely used [42,71,72,82,83,84,85,86,87,88,89,90,91]. The prognostic value of LD is a new concept that can be correlated and related to other variables such as micrometastases, the presence of extracapsular LN infiltration, and the anatomical location of positive lymph nodes [92]. In the future, when the LND range is defined and validated, LD can be used as an eligibility criterion for adjuvant treatment after cystectomy.

Another factor assessed according to the pathological examination of nodal specimens is tumor infiltration outside the LN capsule (ECE) [93]. In 2001, it was established that the presence of this factor indicates a worse prognosis in the population of patients with bladder cancer, and subsequent reports confirmed this relationship [58,82,86,94,95,96,97]. In a 2015 meta-analysis of 1893 patients with LN+ disease, ECE was significantly correlated with reduced relapse-free survival (HR 1.56, 95% CI 1.13–2.14) and cancer-specific (HR 1.60, 95% CI 1.29–1.99) but not overall survival (HR 1.47, 95% CI 0.71–3.05) [98]. Despite these findings, formal recommendations for using these additional prognostic markers are limited due to the lack of prospective data.

Recently, attention has also been focused on the assessment of the size of nodal metastases. In a study assessing the diameter of the largest nodal metastasis after RC, the authors showed that patients with metastases <5 mm had a better median survival (64 vs. 16 months) [96]. A more detailed analysis, also taking into account the percentage of the node volume occupied by neoplastic cells in the area of the largest metastases, did not bring any prognostic implications [99]. The cumulative length of all nodal metastases was also analyzed as a prognostic factor. Stephenson et al. confirmed the prognostic implications of this measurement using a 20 mm cutoff. In multivariate analysis, values below the applied limit were associated with improved overall survival (HR: 1.1; 95% CI: 1.01–1.2; *p* = 0.035), free from relapse (HR: 1.1; 95% CI: 1.03–1.2; *p* = 0.04) and cancer-specific survival (HR: 1.1; 95% CI: 1.03–1.2; *p* = 0.005) [100]. In two subsequent publications, the results obtained were inconclusive, which negatively affects the widespread use of this parameter [48,95].

The local stage (pT) of the bladder tumor, although it is an independent prognostic factor for patients with bladder cancer, was also analyzed in terms of the pN+ feature. Few studies detailed the effect of pT on survival in pN+ patients, but the results indicate that survival changes dramatically when the disease exceeds the bladder wall (>pT2), suggesting that it may be of key prognostic value for the patient regardless of nodal status, in qualification for adjuvant therapy [70,85,87,101,102,103,104].

Using different variants of lymphadenectomy explains the differences in the assessment of the pN, but a large discrepancy is observed even when considering a particular LND template [34]. These differences have been noticed even in situations where the same operator or group of surgeons change the way of sending lymph node packets for pathological evaluation. Meijer et al. prospectively analyzed 174 patients with invasive bladder cancer operated by the same group of surgeons in two different hospitals to find differences in the number of removed nodes. While both groups were comparable in terms of general and oncological characteristics, and the mean number of positive lymph nodes did not differ (1.21 vs. 1.51), there was a significant difference between centers in the mean number of removed nodes (16 vs. 28; *p* < 0.001) [105]. In another publication, the authors noted a change in the mean number of LNs removed from 15 to 20 after implementing the pathological reanalysis procedure, if in the original report the number of removed nodes was <16. Additional analysis did not affect the proportion of node-positive patients in the study group (27.9% vs. 27.2%; *p* = 0.89) [106].

Despite the above data in the literature, formal recommendations regarding the routine use of additional prognostic markers are limited mainly due to the lack of prospective analyses that would unquestionably confirmed the value of each variable.

## 4. Complications of Lymphadenectomy

Radical cystectomy is itself a major surgical procedure with potential complications often related to urinary diversion. Nevertheless, LND and its variants do not appear to have a significant effect on the number of complications, and in some cases, lymphadenectomy facilitates cystectomy [34]. Especially in the case of experienced operators, LND did not increase the perioperative risk, and in fact, the use of anatomical resection technique results in one of the lowest rates of perioperative complications and mortality [87]. The median duration of surgery has been shown to increase by 63 min in extended variant compared to the limited procedure [61,107]. This observation is inconsistent with the results of Jensen et al., who found no significant time difference between the extended and limited LND (306 min vs. 302 min) [44]. The increase in the number of removed nodes did not increase the number of complications and perioperative mortality [70]. Typical events such as lymphocele and lymphoedema were reported in 2% of patients with less than 16 LN removed and in 1% of patients with more than 16 LNs removed [108].

Similarly, expanding the anatomical limits of LND did not increase postoperative complications [42,61,70,96]. In addition, the complication rate requiring additional intervention does not appear to be significantly different between the limited (9%) and the extended variant (11%) [61]. The range of the superextended lymphadenectomy did not significantly affect the number of complications compared to the more limited variants [93,94,109]. These data allow the conclusion that if a patient is fit and healthy enough to undergo radical cystectomy, there are no absolute contraindications for LND, regardless of age and comorbidities.

## 5. Current Guidelines

Despite the growing amount of evidence supporting the use of LND during radical cystectomy in MIBC, there is no clear consensus on the optimal anatomical area of the removed lymphatic tissues and the presentation method for pathological examination. Currently, the most comprehensive and accurate recommendations have been presented by an expert panel of the Joint Société Internationale d’Urologie–International Consultation on Urological Diseases (SIU–ICUD) [110]. The authors recommend a standard lymphadenectomy. Concerning the broader area of lymphadenectomy, the recommendations for its use were limited, referring to the results of the LEA study, which did not demonstrate the advantage of the extended over the standard variant [74]. Moreover, the minimum number of tissue packages was determined: external iliac, internal iliac, and obturator collected on both sides. The minimum number of removed lymph nodes has not been defined, although it has been emphasized a higher probability of detecting at least one positive lymph node with more tissues removed. In the National Comprehensive Cancer Network (NCCN) study, it is recommended that in patients undergoing partial or radical cystectomy, bilateral lymphadenectomy should be performed, covering at least the area of the common iliac, internal iliac, and obturator fossa. In the discussion, the authors also suggest extending the lymphadenectomy to the periaortic and paracaval areas [111].

American Association of Urology (AUA) guidelines include two recommendations for LND. The first one imposes the obligation to perform bilateral dissection during each cystectomy, the intention of which is radical therapy. The second guideline relates to the extent of LND and includes removing at least external and internal iliac areas and obturator fossa region [112]. The Canadian Urological Association guidelines recommend lymphadenectomy for staging and therapeutic purposes in the same scope as proposed in the AUA guidelines. The review presents information about the available results of studies assessing a broader range of LND without formulating an unequivocal guideline [113]. As the only one of the discussed institutions, NICE (National Institute for Health and Care Excellence, UK) has not yet issued clear recommendations regarding this problem. In their guidelines, European Association of Urology experts conservatively proposed only one recommendation that lymphadenectomy must be an integral part of the cystectomy, without specifying the extent of the removed tissues. In the discussion, the authors emphasize that an extended LND template, compared to a limited one, may have a beneficial therapeutic effect; however, the low quality of the available data does not allow us to draw unambiguous conclusions and define clear recommendations [7].

Summarizing, the cited recommendations mostly give quite general references without precisely specifying and standardizing the anatomical area of the lymphadenectomy, which “allows” freedom for the operators. This prompts further work to define the ideal topoanatomical range to maximize oncological outcomes while minimizing the risk of LND-related complications. The lack of prospective studies comparing the areas of lymphadenectomy is one of the reasons for the heterogeneity and noninstantiation in the recommendations in the available guidelines developed by various organizations.

## 6. Minimally Invasive Surgery and Lymphadenectomy

The encouraging results of studies comparing robot-assisted radical cystectomy (RARC) with the open technique have led to RARC being increasingly used. These studies demonstrated the oncological efficacy and surgical feasibility of RARC. Although most RARC studies did not evaluate LND as a primary endpoint, extent of lymphadenectomy and lymph node function are well-defined surrogate markers of surgical quality that may have a direct impact on oncological outcomes [51]. The RAZOR study (randomized open versus robotic cystectomy) was the first open-label, multicenter, phase III study conducted to determine whether RARC was noninferior to open radical cystectomy for progression free survival in two-year observation [114]. The type of surgical intervention did not influence the choice of extended LND (RARC: 51% vs. open: 55.3%, *p* = 0.46), the number of removed lymph nodes (RARC: 23.3 ± 12.5 vs. open: 25, 7 ± 14.5, *p* = 0.13), and pathological nodal stage (N1–RARC: 8.7% vs. open: 5.3%, N2—11.3% vs 8.6%, N3—3.3% vs 2.0%, *p* = 0.55). Additionally, Parekh et al. found no significant differences between the groups in the assessment of postoperative complications related to LND (RARC: 2.0 vs. open: 2.6%). A similar observation was reported by the authors of the phase III study with the primary endpoint of 90-day complications. Clavien grade ≥3 cases in both arms was noted with similar rate (RARC: 21.7% vs. open: 20.7%, *p* = 0.9). Bochner et al. found no difference in the extent of LND, the number of lymph nodes, and the percentage of positive lymph nodes [115]. Moreover, there was no difference in oncological outcomes regardless of the surgical technique [116].

The advantages of the robotic system have resulted in conventional radical laparoscopic cystectomy being largely replaced by RARC for a minimally invasive technique. Khan et al. in the CORAL study reported the long-term early-phase oncological results of a randomized, open (*n* = 20), laparoscopic (*n* = 19), and robotic (*n* = 20) radical cystectomy [117]. All operations were performed by three experienced surgeons, each in the field of appropriate surgical technique. Urinary diversion in minimally invasive study arms were performed by open technique. The mean number of removed lymph nodes was 18.8, 16.3, and 15.5 for open, robotic, and laparoscopic surgery, respectively. The differences in the lymph nodes yields were statistically significant only between the open and the laparoscopic cohorts (*p* = 0.01). Positive lymph nodes were reported in 31.5% (6/19), 30% (6/20), and 15% (3/20), of the laparoscopic, open, and robotic arms, respectively. As in previous studies, the authors did not find a significant difference in the percentage of 90-day complications between groups [118].

Based on this finding, it can be concluded that minimally invasive dissection of the lymph nodes is possible and provides equivalent results compared to the open technique. Experienced surgeons using a robotic or laparoscopic technique can achieve similar lymph node efficiency and complication rates as with the open procedure.

## 7. Future Perspectives

Radical cystectomy (RC) with lymphadenectomy is recognized as the standard of care in the treatment of MIBC (>pT1). It should be emphasized that 25% of RC patients with LND have positive LN in the pathological examination. It is the factor that most strongly correlates with mortality in MIBC. Considering that the diagnosis of nodal invasion based on imaging tests is unreliable, various efforts have been made to increase diagnostic value.

Radiomics is a relatively new in diagnostics field and it is expected to progress rapidly and improve clinical decision-making. This concept utilizes different features from radiological images, which are used to create statistical models to uncover disease characteristics [119]. First radiomics nomogram incorporating the additional measures and CT-reported lymph node status shows favorable predictive accuracy for lymph node metastasis in patients with bladder cancer (AUC—0.9262 (95% CI 0.8657—0.9868)) [120]. Similar nomogram using MRI imaging was constructed with encouraging results both in test and control groups [121]. These new tools have potential for routine use, but effectiveness needs to be validated.

At present, the results of the SWOG-1011 trial are awaited [122]. This similar to the LEA prospective, randomized, controlled phase III study is currently in observation phase, but data are immature. This trial compares the oncological outcomes between standard (S-LND) and superextended (SE-LND) lymphadenectomy. The study includes 650 patients, and the SE-LND group achieved a 10% higher three-year disease-free survival than S-LND (65% vs. 55%). One of the most notable differences is the exclusion of pT1 patients. Additionally, neoadjuvant and adjuvant chemotherapy were used in the therapy. However, it should be remembered that perioperative systemic treatment may affect the research results, reducing the reliability of the survival benefits of LND alone. Other endpoints are estimation of operative time depending on the type of surgery (with or without nerve sparing, assessment of intraoperative, perioperative, and 90-day mortality), length of hospital stay, and local and retroperitoneal recurrences within soft tissues in patients assigned to perform S-LND compared to E-LND. In addition, blood samples were analyzed for the presence of circulating tumor cells (CTCs) and markers of epithelial and mesenchymal transition and their correlation with primary tumor stage and lymph node metastases.

Radio-guided surgery using radiotracers or fluorescent markers have potential for use as a diagnostic tool for individualized pre- and intraoperative detection of lymph node metastasis in BCa. In a study conducted by Rosenblatt et al., the LN+ detection rate was achieved at the level of 92%, using technetium injected peritumorally [123]. However, the use of a similar method in the studies by Aljabery et al., was characterized by a sensitivity of only 67% [124]. A series provided by Polom et al. showed the specificity of using the combination of technetium radiocolloid and indocyanine green at the level of 47% [125]. The results should be interpreted with caution due to the small sample size in previous studies and require verification on larger cohorts.

The use of new cell cycle checkpoint inhibitors, which in the tumor microenvironment enable tumor antigen-specific stimulation of cytotoxic T lymphocytes to destroy neoplastic tissues selectively, seems very promising at present, which also brings benefits in the treatment of LN+ patients [126,127,128].

## 8. Conclusions

An integral part of the RC in BCa is bilateral lymphadenectomy. It enables the precise determination of the N stage, the need for adjuvant treatment, and remains the best therapeutic form for LN+. Many compatible and consistent observations indicate that extended lymphadenectomy impacts disease-free survival, regardless of the baseline degree of LN invasion and stage of BCa, additionally providing an oncological benefit over standard template. However, the quality of available data is low, which implies the need for further work on cross-sectional, randomized, prospective analyses. Evidence-based standardization of the lymphadenectomy procedure required to establish surgical guidelines is essential to improving the quality of procedures and patient survival.

## Figures and Tables

**Figure 1 medicina-57-00415-f001:**
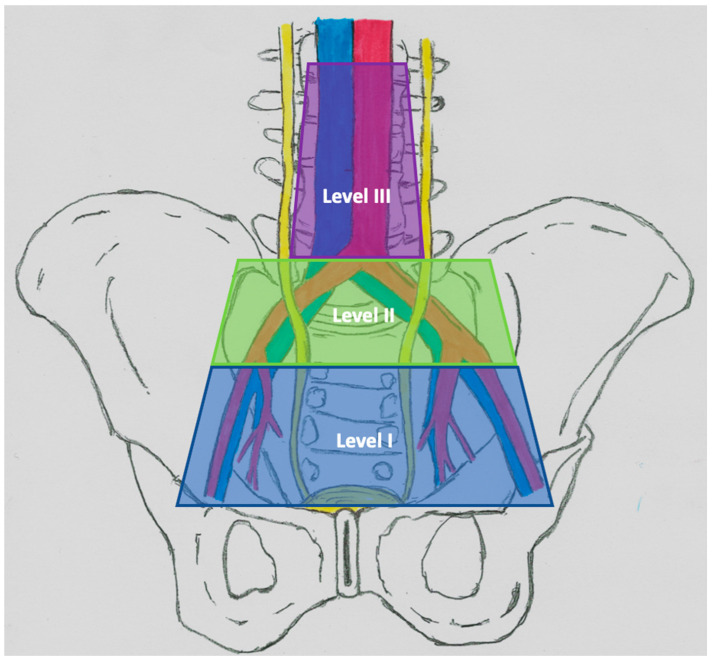
Anatomical range of lymphadenectomy corresponding to the resection level proposed by Leissner et al. (explanatory notes) [42].

**Figure 2 medicina-57-00415-f002:**
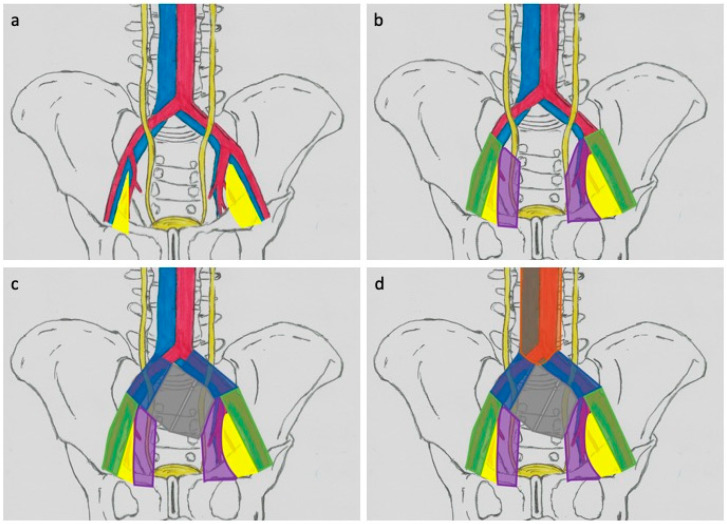
Anatomical diagram of lymphadenectomy divided into ranges: (**a**)—limited, (**b**)—standard, (**c**)—extended, and (**d**)—superextended (explanatory notes in the text); obturator area (yellow), external illiac vessels (green), internal illiac vessels (purple), common illiac vessels (navy), and presacral area (grey).

## Data Availability

Not applicable.

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
