# Peer review of "The Usefulness of Lymphadenectomy in Bladder Cancer—Current Status"

_medicina, 2021, doi:10.3390/medicina57050415_

Round 1
Reviewer 1 Report
This is a well-written review paper on the role of lymph node dissection (LND) in non-metastatic bladder cancer patients. The authors comprehensively review the diagnostic and prognostic roles of LND in such patients. The authors may want to address the following points before publication. 1_ Line 134: A term “trigone” may be general instead of “triangle”. 2_ Line 280: What did the authors mean “specific survival”? Please specify it.Author Response
Thank you very much for taking the time to review the manuscript and for providing valuable feedback. The issues you had mentioned have been analyzed and fixed.
We have made following changes in the main manuscript according to your remarks and highlighted it in yellow.
1_ Line 134: A term “trigone” may be general instead of “triangle”.
- term was changed for "trigone"
2_ Line 280: What did the authors mean “specific survival”? Please specify it.
- the quoted results refer to overall survival - we made a correction and instead of the word "specific", there is "overall" - which clears up any inconsistencies for the readers
In addition, an English native speaker has reviewed the manuscript to ensure there are not grammatical mistakes.
Your time spent to review our manuscript and the feedback you have provided is very much appreciated! Thank you again, for helping us improve our submitted paper.
Sincerely yours,
Bartosz Małkiewicz and Paweł Kiełb
Reviewer 2 Report
This work is a narrative review regarding lymphadenectomy during radical cystectomy for bladder cancer. The authors gathered data regarding detection of node-positive disease, the impact of lemphadenectomy in patient survival, and the extent of lymphadenectomy as suggested by different study groups.
The authors mainly ruminate old knowledge, and selected references are outdated. They do not present the impact of new technology (eg robotic surgery) to the outcomes of LN dissection. Furthermore, they do not give future perspectives to boost further research.
Author Response
We would like to thank you for reviewing our manuscript and for comment of our work.
We have considered all remarks and modified the manuscript based on your suggestions.
All the changes have been highlighted in yellow through the manuscript. We feel that the quality of the manuscript has been significantly improved with these modifications.
We strongly agree with the Reviewer regarding the lack of data on the use of minimally invasive technologies in the work. Therefore, we have added a subsection on new technologies and their impact on lymphadenectomy (6. Minimally invasive surgery and lymphadenectomy), which should positively affect the comprehensiveness of the article.
With regard to the selection of references, we also agree with the Reviewer that indeed the publication dates of some of the cited articles are not the latest, but their results remain valid and up to date. We want to emphasize that the in-depth analysis of the topic and the cross-sectional nature of the work required citing the results of fundamental and milestones in this topic, and these were published more than 5 years ago. But a selection of the literature reveals that lymphadenectomy issues have been difficult to solve over the years. This, in consequence, affects the current state of knowledge in this topic. And that's how it was presented in this narrative review. The remaining references have been published over the past few years and present the current achievements.
In accordance with the Reviewer's suggestion, we have introduced changes to the paragraph regarding future perspectives to take into account the latest research directions aimed at solving problems related to lymphadenectomy
In addition, the manuscript proofreading was done by a native English speaker to make sure there were no grammatical errors.
Your time spent to review our manuscript and the feedback you have provided is very much appreciated! Thank you again, for helping us improve our submitted paper.
Sincerely yours,
Bartosz Małkiewicz and Paweł Kiełb
Round 2
Reviewer 2 Report
No further comments